# Autophagy-Related MicroRNA: Tumor miR-125b and Thyroid Cancers

**DOI:** 10.3390/genes14030685

**Published:** 2023-03-09

**Authors:** Liudmila V. Spirina, Irina V. Kovaleva, Svetlana Yu. Chizhevskaya, Anastasiya V. Chebodaeva, Nataliya V. Tarasenko

**Affiliations:** 1Ministry of Health of Russia, Siberian State Medical University, 634050 Tomsk, Russia; 2Cancer Research Institute, Tomsk National Research Medical Center of the Russian Academy of Sciences, 634050 Tomsk, Russia; 3Research Institute of Medical Genetics, Tomsk National Research Medical Center of the Russian Academy of Sciences, 634050 Tomsk, Russia

**Keywords:** papillary thyroid cancer, autophagy, hsa-miR-125b, tumor invasion, BRAFV600E, risk of recurrence

## Abstract

Background: Autophagy is a stress response mechanism that causes cellular components to degrade. Its defects were associated with multiple pathologies, including cancers. Thyroid cancer is known to be the most prevalent form of malignant neoplasm among endocrine tumors. The aim of the study was to seek and comprehensively explore the role of autophagy related genes and proteins play in thyroid cancers through bioinformatics analysis with their detection in the tissue samples. Methods: Bioinformatics analysis was performed to investigate autophagy related proteins and genes involvement in thyroid cancer progression. The experimental verification was done in cancer samples of one hundred and three patients with thyroid pathology included in the study. The miR-125blevel was detected by PCR in real time. Results and discussion: The bioinformatics analysis verified the miR-125b as a regulatory mechanism in autophagy. Its expression in patients with PTC was reduced by 6.75 times in cancer patients compared to the patients with benign tumors. The BRAFV600E mutations were associated with a decrease in hsa-miR-125b expression by 12.67 times compared to tumors with the wild-type gene. Conclusions: Our findings revealed involvement of the autophagy related proteins in cancer progression. The significant mechanisms of regulation are non-coding RNA sequences implicated in a variety of oncogenic processes. We found that miR-125b is a potential maker in thyroid cancer invasion, BRAV600E mutational status and risk of recurrence.

## 1. Introduction

Autophagy is a stress response mechanism that leads to the degradation of cellular components, the defects of which are associated with multiple conditions [1]. Mounting evidence reveals that autophagy is a factor in thyroid cancer. Modulating autophagy has become a prominent target for improving the monitoring of cancer patients [2]. The principal triggers that determine the induction of autophagy are starvation, reactive oxygen species, endoplasmic stress and hypoxia. There are multiple channels that modulate autophagy, such as AMPK and PI3K orchestrating the protein cascade (ULK1, ATG16L1, MAP1LC3B, PRKAA1, ATG12, ATG14, ATG5, mTOR and BECN1) [2].

Papillary thyroid cancer (PTC) is the most frequent differentiated tumor. The malignant neoplasms rate in the thyroid gland is increasing annually worldwide [3]. The molecular mechanisms underlying the progress of PTC are complex and diversified. A number of key mutations are associated with PTC progression, including tyrosine kinase RET1 translocation, RAF protein mutation (BRAFV600E) and telomerase mutation (TERT1) [4]. However, their role in PTC is relatively low compared to the multiplication of oncogenic signals such as epigenetic factors.

Differentiated thyroid cancer (DTC) shows a good prognosis. The MAPK (mitogen-activated protein kinase) and AKT/mTOR signaling pathways are the main molecular events in the aggressive behavior of cancer. Promising approaches with a variety of targets have been found to restore autophagy in preclinical studies. But the complete molecular mechanism underlying the poor prognosis based on autophagy impairments remain fuzzy [1].

Recent studies have revealed the role of non-coding RNA sequences, particularly microRNAs, in a variety of pathologies. They are important non-coding endogenous RNA sequences of 22 long nucleotides that can regulate up to 30% of all human genes [5]. Small RNAs are involved in the most cellular functions [2]. Autophagy-regulating miRNA studied in cancer is a challenge in malignancies understanding and in the search for anti-cancer therapy targets.

Enhanced expression of p-ULK1 may promote autophagy by phosphorylating Beclin-1 and activating lipid kinase VPS34 [6]. The 16L1 autophagy-related protein (ATG16L1) indicates the process of autophagosome creation as well as the membrane closure process [7]. The light chain 3B protein associated with microtubule 1A/1B (hereinafter referred to as LC3) is a protein that in humans is encoded by the gene MAP1LC3B. It is a core protein in the autophagy pathway [8].

Targets of rapamycin (TOR) kinase-containing protein complexes are main regulators of the autophagic pathway. The mammalian TOR (mTOR) kinase forms two autophagic protein complexes: mTORC1 and mTORC2 [7]. Various stress-causing signals, deprivation and energy deficiencies are induced by the AKT/mTOR and the AMPK pathway. It orchestrates cell growth-related events and protein biosynthesis. The process is initiated by the Class III phosphatidylinositol 3-kinase complex (PI3K), comprising the proteins PI3K VPS34 and VPS30, ATG14/Barkor, VPS15 and ATG6/BECN1 (Beclin 1). The lipid kinase activity of the PI3K complex is responsible for the accumulation of phosphatidylinositol molecules 3-phosphate (PI3P) on the membranes, including the external foliole of the endoplasmic reticulum (RE). Two ubiquitylation-type conjugation systems, ATG12-ATG5-ATG16 and ATG8 (MAP1LC3, or briefly LC3 in mammals), regulate the elongation and completion of autophagic membranes [9].

MicroRNAs are considered novel tools in autophagy modulation. Several ones were associated with the change in autophagy-related proteins that are playing a role at various steps of the autophagy flux [6].

Hsa-miR-125b is derived from miR-125b-1 and miR-125b-2. In turn, miR-125b-1 was obtained from a long non-coding RNA (lncRNA)–MIR100HG (miR-100/let-7a-2/miR-125b-1, chromosome 11), and miR-125b-2 was obtained from the miRNA cluster (miR -99a/let-7c/miR-125b-2, chromosome 21). miR-125 is a highly conserved miRNA throughout diverse species. It consists of three homologs: hsa-miR-125a, hsa-miR-125b-1 and hsa-miR-125-2. miR-125 targets a number of genes such as transcriptional, growth factors and various components of signaling cascades [10].

Dysregulation of hsa-miR-125b is associated with tumor progression. It is known that small non-coding sequences of RNA can be presented both as oncogenes and tumor suppressors [11]. The hsa-miR-125b role in the oncogenes is still unclear. It was shown that it reduced the PI3K, phospho-Akt, and phospho-mTOR expression in anaplastic thyroid cancer cells [12]. In addition, it was found that miR-125b affects glucose metabolism by changing the proliferative activity of the tumor [13] and also affects the growth factor receptors [14] and transcription factors (NF-kB) [15], determining the metastatic and invasive cancer potential [16]. An association between the miRNA’s rate and the BRAFV600E mutational status was found. It is a significant indicator of the aggressive PTC phenotype [17].

Wang et al. (2018) demonstrated a new mechanism in hsa-miR-125b influence on the Foxp3 transcriptional factor. This factor is able to stimulate autophagy, modifying the sensitivity to chemotherapy [16,18]. Autophagy is a unique self-degrading process affecting cancer progression and initiation.

We aimed to seek and comprehensively explore the role of autophagy related genes and proteins play in thyroid cancers through bioinformatics analysis with their detection in the tissue samples.

## 2. Materials and Methods

### 2.1. Protein-Protein Interaction (PPI) Analysis

The Search Tool for the Retrieval of Interacting Genes/Proteins (STRING) used to review all known and predicted associations between proteins. The interacting network of autophagy-related proteins with other associated proteins were visualized through the STRING database. The cutoff value of interaction score was set to 0.15. A Venn diagram was utilized to display the interactions.

### 2.2. MicroRNA and Genes Associations

For pathway enrichment analysis, both Kyoto Encyclopedia of Genes and Genomes (KEGG) pathway analysis and Gene Ontology (GO) enrichment analysis was performed to visualize the cellular pathways. The cutoff Q value and P value were set at 0.05 and 0.01, respectively. The DIANA miRpath and PANTHER databases were used for bioinformatics analysis. HumanTargetScan (ver. 8.0) and miRTarBase (ver. 9.0) were used to seek target genes for miR-125. To identify the transcription factors, Transmir base (version 2.0) was used (https://www.cuilab.cn/transmir; accessed on 27 January 2023).

### 2.3. Clinical and Experimental Tools

The study included one hundred and three patients with thyroid pathology treated at the Cancer Research Institute of the Tomsk National Research Medical Center: sixty-seven patients had PTC and thirty-six patients had benign thyroid pathology. The T1-2N0M0 stage detected in twenty-seven patients and T3-4N0-1M0 was detected in forty patients. Metastases in regional lymph nodes were presented in twenty-seven patients. According to the histological subtype of PTC, patients were divided into a follicular subtype group (12 patients) and a classical subtype group (55 patients). The BRAFV600E mutation was identified in 18 patients. According to the criteria of the American Thyroid Association (ATA, American Thyroid Association), patients were divided into recurrence risk groups as follows: low risk of thyroid cancer recurrence was observed in 23 patients, intermediate risk in 25 patients and high risk in 19 patients.

This work was approved by the Local Ethical Committee in the Cancer Research Institute of the Tomsk National Research Medical Center. All procedures were carried out in accordance with the Protocol of the Helsinki Declaration on Human Rights (1964). All patients included in the study signed an informed consent to participate in the study.

The material of the study was samples of tumor and unchanged tissue obtained during surgery. Samples were taken in points located at a distance of at least 1 cm from the tumor boundaries. The samples were frozen and stored at a temperature of −80 °C.

### 2.4. DNA Extraction

DNA was isolated using the FFPET DNA Extraction Kit (Biolink, Russia). To assess the amount of isolated DNA, its concentration was assessed on a NanoDrop-2000 spectrophotometer (Thermo Scientific, Waltham, MA, USA). The resulting DNA was used for real-time PCR.

### 2.5. BRAFV600E Mutation Analysis

The BRAFV600E mutation was determined using the Real-time-PCR-BRAF-V600E kit (Biolink, Novosibirsk, Russia), designed to detect the GTG→GGG point mutation in codon 600 of the BRAF gene. The analysis was carried out by real-time allele-specific PCR.

### 2.6. Quantitative PCR with Reverse Transcription in Real Time

The tumor samples were incubated in RNAlater solution (Ambion, Foster City, CA, USA) for 24 h at 4 °C and then stored at −80 °C. MicroRNA was extracted using an isolation Kit (Biolabmix, Novosibirsk, Russia). The quality of the isolated nucleic acids was carried out using capillary electrophoresis on a TapeStation device (Agilient Technologies, Santa Clara, CA, USA). The RIN ranged from 2.2–3.3.

RT-qPCR was performed according to [Pfaffl MW]. PCR was conducted in 25 μL reaction volumes containing 12.5 μL BioMaster HS-qPCR SYBR Blue (2×) (“Biolabmix” Russia) and 300 nanoM of each primer: hsa-miR-125b: Forward primer 5′-GGATTCCCTGAGACCCTAAC-3′, Reverse primer 5′-GTGCAGGGTCCGAGGT-3′, RT primer 5′GTCGTATCCAGTGTCAGGGTCCGAGGTATTCGCACTGGATACGACTCACAAG-3′; U6: Forward primer 5′-CTCGCTT CGGCAGCACATATACT-3′, Reverse primer 5′-ACGCTTCACGAATTTGCGTGTC-3′, RT primer 5′-AAAATATGGAACGCTTC ACGAATTTGG-3′. A pre-incubation at 95 °C for 10 min was performed to activate the Hot Start DNA polymerase and denature the DNA and was followed by 45 amplification cycles of 95 °C denaturation at 95 °C for 10 s, 60 °C annealing at 60 °C for 20 s (iCycler iQ™, BioRad, Berkeley, CA, USA). The fold changes were calculated by the ΔΔCt method (the total ΔΔCt = fold of cancerous/normal tissue gene level), using normal tissue. The RNU6 was used as the endogenous control.

Statistical analysis was performed using the Statistica 12.0 software package (Tulsa, OK, USA). Normal distribution check was conducted using the Kolmogorov-Smirnov test. The results of gene expression determination are presented as Me (Q1; Q3). The Mann-Whitney test was used to assess significant differences. Differences were considered significant when *p* < 0.05.

## 3. Results

### 3.1. Investigation of MicroRNAs Related to Autophagy

#### 3.1.1. Investigation of MicroRNAs Regulating Proteins Associated with Autophagy

Autophagy is a universal process of cellular viability as well as cancer cells. The key autophagy associated proteins are used to study the STRING database: ULK1, ATG16L1, MAP1LC3B, PRKAA1, ATG12, ATG14, ATG5, mTOR and BECN1 (Figure 1). The analysis showed the following associations between autophagy-associated proteins with multiple links. The edges colored purple reflect experimentally proven relationships, blue—relationships, information about which is obtained from supervised databases. Green edges reflect the neighborhood in the genome, blue—the joint occurrence, red—the fusion of genes. Light green ribs mean a joint mention of these proteins in Pub-Med Abstracts, black—co-expression, and light blue—homology. The found data highlighted the multiple interactions between the proteins.

The bioinformatics analysis in the DIANA microRNA database identified microRNAs regulating proteins associated with autophagy (Appendix A). Twenty-two microRNAs were found for ATG12. ATG14 was regulated by the twenty-three microRNAs, ATG16L1 by seventy-five microRNAs, ATG5 by twenty-two microRNAs, BECN1 by forty microRNAs, MAP1LC3 by thirty microRNAs, ULK1 by one hundred six microRNAs, mTOR by thirty-three microRNAs, and PRKAA1 by two hundred sixteen microRNAs.

#### 3.1.2. MicroRNAs Associated with the Thyroid Cancers

At the next step, data on the microRNAs associated with the thyroid cancer and the proteins associated with autophagy were analyzed. Twenty-three microRNAs were identified with upregulated-expression and thirty-nine had oncosuppressor activity, which regulate all stages of oncogenesis (Table 1). However, among the targets associated with autophagy, only one microRNA is shown. Thus, miR-125b are able to regulate transcription and translation of the ATG5 protein, mediating the resistance to therapy in papillary thyroid cancer [19].

#### 3.1.3. Targets and Signaling Cascades Associated with MicroRNA 125

A bioinformatic analysis was performed to look for potential target genes and protein-protein interactions between their products (Table 2). Forty-four genes were found to be targets for the miR-125b. The detailed review showed the participation of the miR-125b in multiple processes: cell motility, immune response, DNA methylation, growth and transcriptional factors signal transduction. The MAPK and AKT/mTOR signaling cascades are found to be the key ones in autophagy [63]. PIK3CA, the initial component of the AKT/mTOR cascade [64]. CCDC6, protein 6 containing an antenna domain, interacts with CREB1 (reaction element binding protein cAMP 1) and suppresses its transcriptional activity [65].

The STRING database was used for the creation of the protein-protein interactions between the proteins of miR-125b associated genes (Figure 2). The complex scheme recovers only four target proteins that are involved in thyroid cancer progression.

KEGG pathway enrichment analysis revealed microRNA targets. They are components of the following signaling pathways: ECM-receptor interaction, proteoglycans in cancer, lysine degradation, adherent junction, viral carcinogenesis, thyroid cancer, focal adhesion, one carbon pool by folate, TGF-beta signaling pathway, colorectal cancers pathway (APC and Lynch syndrome), central carbon metabolism in cancer and FoxO signaling pathway. Of particular interest are the components involved in the molecular mechanisms of thyroid cancer development. Its main components are listed in Table 3.

The ECM-receptor interaction is essential in the cancer heterogeneity investigation [66]. Proteoglycans are molecules, responsible for cellular behavior modulation [67]. Cadherins and catenins are the central cell-cell adhesion molecules in adherens junctions (AJs). The significance of cadherin and catenin in cancer initiation and progression is well known [68]. The additional pathogenic mechanism is Lysine-specific demethylase 1 (LSD1) that stabilizes hypoxia-inducible factor 1α (HIF1α) to promote tumor progression [69]. The associations between viruses and cancer have been conducted in several studies and verified in meta-analyses [70]. Molecular alterations underlying PTC progression include deregulation of focal adhesion kinase (FAK), and PTC progression correlates with mRNA FAK-Del33 and pY397-FAK [71].

KEGG analysis revealed that one carbon pool by folate was connected to PTC pathogenesis [72]. A study by Xie et al. suggested that TGF-β1 triggers invasion and migration of PTC cells by inhibiting the expression of lncRNA-NEF [73].

We revealed gene and signaling components responsible for PTC manifestation. Lynch syndrome describing a familial cancer syndrome with four DNA mismatch repair genes mutations was found to be associated with PTC [74]. A high risk of familial adenomatous polyposis (FAP) was found in PTC patients [75].

PTC exhibits impaired energy metabolism, abnormal cell division, and more aggressive behavior through metabolic reprogramming and changes in the central carbon metabolism [76]. The FOXo signaling cascade is the important regulator of oncogenesis, which involves the mechanism of the RBM47/SNHG5/FOXO3 axis in PTC and also affects the triggering of autophagy [77].

The association of microRNA 125b with transcription factors is shown in Figure 3. The association of hsa-mir-125b with STAT3, PRDM1, TP53, EZH2, NFkB1, AKT1, ERS1 and PPARG was revealed.

STAT3 transcriptional factor regulates PTC metastasis via glycolysis modification [78]. PRDM1 is a key target in Hashimoto thyroiditis [79]. TP53 is an oncosuppressor associated with metastatic PTC [80].

Recent studies revealed the EZH2 gene (histone methyltransferase) in modulating the cancer invasiveness and metastatic potential [81]. NF-κB induction is a key process in cancer progression and promotes NFKB1 [82]. The somatic mutations were identified in AKT genes and is associated with PTC risk [83].

Thyroid cancers are prevalent in women, indicating the sex hormone disorders as an essential risk factor [84]. The peroxisome proliferator receptor gamma (PPARγ) alteration associated with steroids transcriptional factors was detected only in follicular thyroid cancer and follicular adenoma [85].

### 3.2. MicroRNA 125b Expression in Thyroid Pathologies

The miRNA expression was reduced by 6.75 times in patients with PTC compared with the benign tumors. miRNAs are found to play a multifaceted role in oncogenesis. It is believed that they could be both oncoproteins and oncosuppressors, guiding the cancer progression potential [11].

Table 4 presents data on the microRNA expression in PTC. Thus, tumor size and regional lymph nodes involvement were not associated with the hsa-miR-125b level. However, in patients with the thyroid gland capsule invasion, hsa-miR-125-b expression increased 2.36 times compared to patients without signs of invasion.

The histological subtype of thyroid cancers was associated with miR-125b expression. A decrease of 37.0 times was found in the follicular subtype compared to tumors with the classical subtype. Additionally, in patients with positive BRAF V600E mutational status, a significant decrease in hsa-miR-125b levels by 12.67 times was shown compared to tumors without the mutation.

To assess the miR-125b contribution to the PTC relapse prediction, we studied the miRNA expression in groups according to the criteria of the American Thyroid Association (ATA, American Thyroid Association). In patients with an intermediate risk of recurrence, the miR 125-b expression was significantly increased by 107.09 and 11.78 times compared with the low- and high-risk groups, respetively.

## 4. Discussion

Autophagy is a universal process of the cellular viability as well as cancer cells. For nine key autophagy associated proteins (ULK1, ATG16L1, MAP1LC3B, PRKAA1, ATG12, ATG 14, ATG5, mTOR and BECN1), we found five hundred sixty-two microRNAs (in Appendix A). Among them only sixty-nine are found to be correlated with PTC. However, among the targets associated with autophagy, only miR-125b is able to regulate transcription and translation of the ATG5 protein, mediating the resistance to therapy in PTC [19].

Non-coding RNA sequences, such as miRNAs, are essential for the autophagic flux modification, from upstream signaling pathways to later stages of autophagy. We found the theoretical and experimental verification of hsa-miR-125b implication in thyroid cancer progression. The revealed forty-four genes were found to be targets for miR-125b. KEEG data analysis and the DIANA microRNA investigation tool show that a variety of genes and transcriptional factors guided the PTC progression. But only few of them are under consideration in PTC research. We indicated four target proteins associated with PTC. They belong to the two signaling cascades that contribute to the cancer progression. MAPK1 and MAPK3 are the components of the rout that orchestrate the rapid cancer cell division and determine the aggressive cancer cells behavior [63]. PIK3CA activates AKT/mTOR signaling and operates the protein’s metabolism as a key mechanism of cells adaptation and survival [64]. The novel genetic marker is CCDC6, the components of the gene’s translocation in PTC [65]. It should be noted that the role of most molecular and genetic factors is still unknown, which causes the unpredictable pattern of behavior of tumor cells and leads to the low treatment response.

MiR-125b targets a number of genes such as components of signaling cascades and transcriptional factors [10]. Most of them are universal and highlight the variety of oncogenic processes: cell motility [66,67,68,69,70,71], epigenetic regulation [69] as well as metabolic reprogramming [72,76], viruses [70], immune response [73] and familial genetic disorders [74,75]. It was again mentioned that the FOXo signaling cascade participation in PTC oncogenesis as well as in autophagy initiation [77].

Transcriptional factor activation is the hidden inner mechanism in cancer progression. The association of hsa-mir-125b with STAT3, PRDM1, TP53, EZH2, NFkB1, AKT1, ERS1 and PPARG was revealed. The complex of factors represented the processes activated in PTC development [78,79,80,81,82,83,84,85]. They recover the mechanism of immune response, microenvironment modification, hormone receptors signal transduction, oxidative stress and etc.

Currently, the diagnostic application of microRNAs in human tumor diseases, as well as for recurrence detection after therapy, is widely discussed. Multidirectional changes in the miR-125b level in the PTC were revealed. The presented data indicate the miRNAs involvement in oncogenesis. There was a decrease in the hsa-miR-125b rate found in patients with PTC compared with benign pathology tissues. But the PTC classical subtype was associated with an increased level of expression.

It is known that the tumor invasion is the cancer progression, associated with the intracellular signaling, transcription and growth factors changes [12,37,65] and reflected in the aggressive tumor behavior. In addition, a high hsa-miR-125b expression was associated with the BRAF V600E status, which was disclosed in the results of the experiment on melanoma in cell culture [17]. The study showed an association between the cancer mutational status and hsa-miR-125b expression.

The tumor invasive potential may be associated with the autophagy, modulating the transformed cell behavior [18]. Activation of “cell self-digestion” is significant in the PTC progression, including the therapy effectiveness. It was found that miRNA expression differs depending on the disease recurrence risk. The maximum values of the expression were recorded for patients with an intermediate risk, which is probably of decisive importance, since tumors with an intermediate risk of relapses may have biological features associated with an aggressive phenotype. In previous research, the LC3B protein content, an autophagosome protein, was noted in this group of patients [86]. Some components of the autophagic machinery have important non-autophagic cellular functions. High MAP1LC3B levels could result from cancer progression and reflect the low sensitivity of anti-cancer therapy [87].

## 5. Conclusions

Five hundred and sixty-seven miRNAs associated with nine autophagy proteins (ULK1, ATG16L1, MAP1LC3B, PRKAA1, ATG12, ATG 14, ATG5, mTOR and BECN1) were identified, of which sixty-two are accompanied by thyroid cancer, and only miR-125b is associated with the autophagy modulation in this pathology.

The involvement of the non-coding RNA sequences remains unclear. It found the multiple genes and transcription factors used in oncogenesis. The concept of the paper was to find the pool of miRNAs associated with autophagy, and then identify the most powerful ones that regulate the oncogenesis in the thyroid gland. The interesting trend was again verified for the thyroid cancers, that determination of one isolated single pathway and route does not explain and predict cancer behavior. There should be found the molecular factors combination that are essential for the oncogenesis, providing new prominent targets in search for anti-cancer therapy.

Autophagy is a universal, powerful tool for thyroid cancer progression. Our findings revealed the hsa-miR-125b expression changes in thyroid cancer invasion, in BRAF V600E mutant tumors and in intermediate recurrence risk. High expression of hsa-miR-125b is a feature of the classical tumor subtype, as well as for tumors with invasion into the thyroid gland capsule. The data obtained indicate the role of miRNAs in the formation of the tumor invasive properties. The role of miRNAs is planned to be investigated further by a more comprehensive study.

## Figures and Tables

**Figure 1 genes-14-00685-f001:**
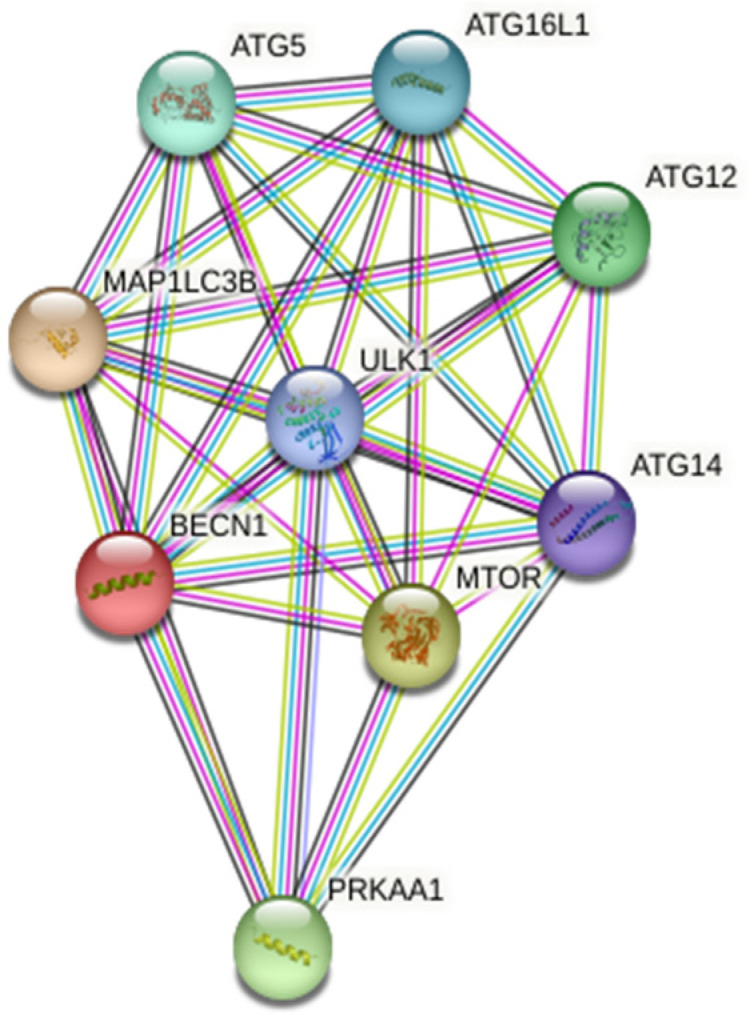
Protein-protein interactions between proteins associated with autophagy (https://string-db.org/; accessed on 27 January 2023). The edges colored purple reflect experimentally proven relationships, blue—relationships, information about which is obtained from supervised databases. Green edges reflect the neighborhood in the genome, blue—the joint occurrence, red—the fusion of genes. Light green ribs mean a joint mention of these proteins in Pub-Med Abstracts, black—co-expression, and light blue—homology. Note: Interactions: 
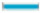
—from verified databases; 
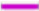
—experimentally determined interactions. Predicted interactions: 
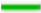
—neighborhood in the genome; 
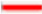
—gene fusion; 
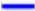
—occurrence. Other: 
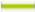
—joint mention of these proteins in PubMed Abstract; 
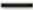
—co-expression; 

—homology.

**Figure 2 genes-14-00685-f002:**
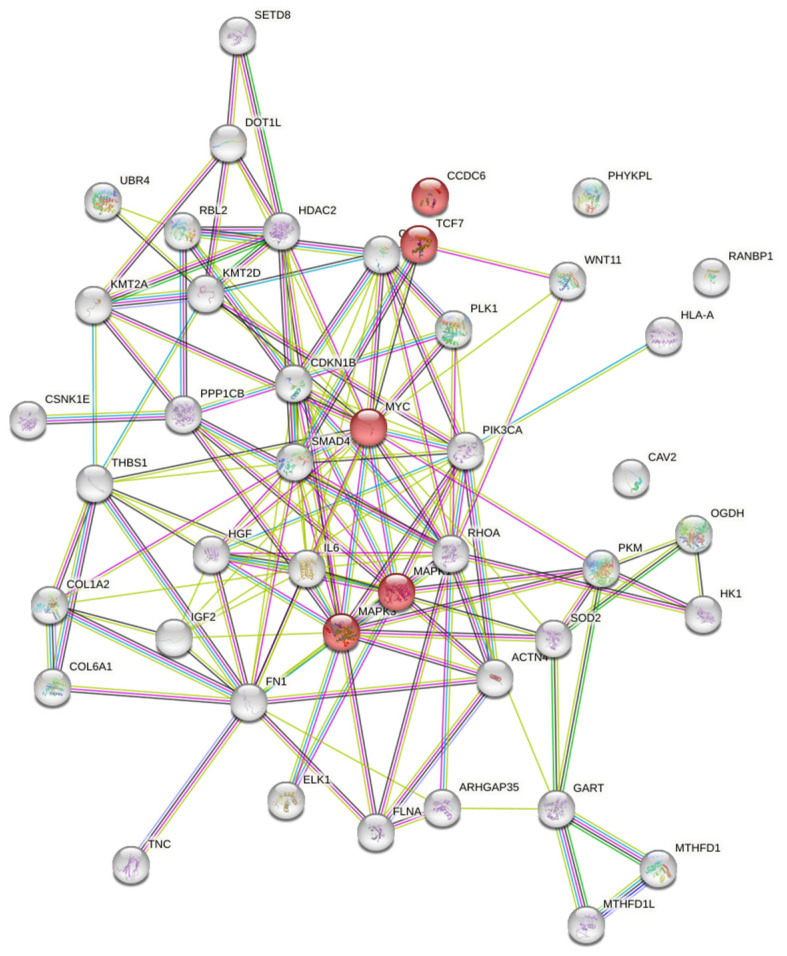
Protein-protein interactions between miR-125b target gene proteins (https://string-db.org/, accessed on 27 January 2023). Note: proteins associated with thyroid cancer are marked in red. Interactions: 
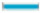
—from verified databases; 
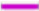
—experimentally determined interactions. Predicted interactions: 
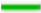
—neighborhood in the genome; 
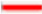
—gene fusion; 
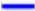
—occurrence. Other: 
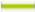
—joint mention of these proteins in PubMed Abstract; 
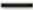
—co-expression; 

—homology.

**Figure 3 genes-14-00685-f003:**
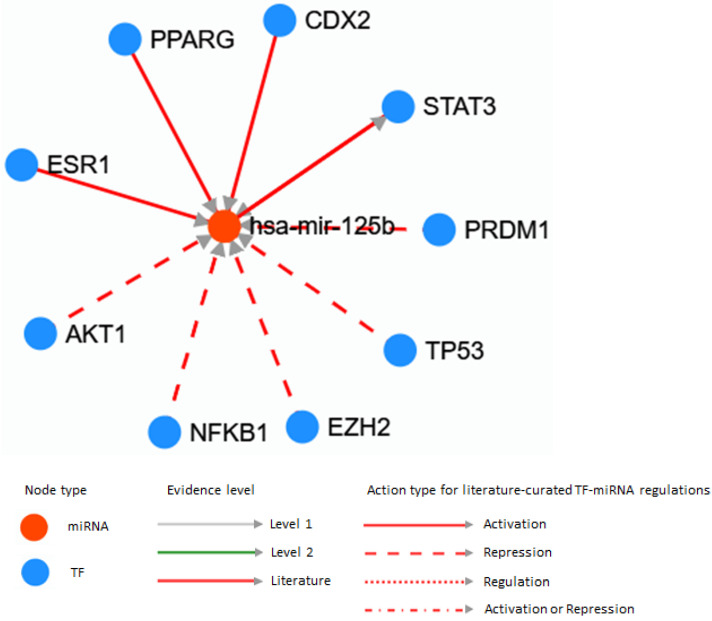
miR-125b and transcription factors associations (https://www.cuilab.cn/transmir; accessed on 27 January 2023). Note: level 1—regulation of TF-miRNA is associated with the analysis of a large number of microRNAs (theoretical connection); level 2—regulation of TF-miRNA is associated with the analysis of experimental data.

**Table 1 genes-14-00685-t001:** List of microRNAs associated with thyroid cancer.

Change in Expression	MicroRNAs (Target/Regulator)	Reference
microRNAs upregulated in thyroid cancers	hsa-miR-146a-5p (TRAF and PML)	[20]
	hsa-miR-146b-5p (RARB)	[21]
	hsa-miR-146b-5p (DICER1)	[22]
	hsa-miR-221 (AXIN, BCL2, RUNX1, CCNE2)	[23]
	hsa-miR-221-3p	[24]
	let-7p	[25]
	miR-144 (AXIN, BCL2, RUNX1, CCNE2, mTOR)	[23,26]
	miR-146a/miR-146b (IRAK1)	[27]
	miR-146b	[28]
	miR-155	[29]
	miR-17-5p	[22]
	miR-181a (RB1)	[30]
	miR-181b	[29]
	miR-182 (TRIM8)	[31]
	miR-19a (PTEN, TSHr, Tg, TTF1 and Pax8, CDH1, an E-cadherin)	[32]
	miR-205 (VERG-A, ZEB1)	[33]
	miR-221 (RECK, TIMP3)	[34,35,36]
	miR-223 (APQ-1)	[37]
	miR-23a (PTEN)	[38]
	miR-340 (BMP4)	[39]
	miR-34a (AXIN, BCL2, RUNX1, CCNE2, AXL)	[23,26]
	miR-451a	[22]
	miR-9-3p (BLCAP)	[40]
microRNA as oncosupressor	miR-132-3p	[22]
	let-7a	[41]
	let-7b (HMGA2)	[41]
	miR-101	[36]
	miR-1179	[37]
	miR-125b (Foxo3, ATG5)	[24]
	miR-128 (SPHK1, Bmi-1, EGFR and E2 F3)	[42]
	miR-129 (MAL2)	[43]
	miR-130a	[12]
	miR-132 (FOXA2)	[44]
	miR-139b-5p (RICTOR, SMAD2/3 and HNRNPF)	[45]
	miR-144	[36]
	miR-144 (E2F8)	[46]
	miR-146a-5p	[22]
	miR-146b	[36]
	miR-148a	[24]
	miR-153-3p (RPS6KB1)	[47]
	miR-181b	[24]
	miR-194	[24]
	miR-199a-3p	[48]
	miR-203 (Survivin)	[49]
	miR-205 (VERG-A, ZEB1, YAP1)	[50]
	miR-21	[51]
	miR-212 (SIRT1)	[52]
	miR-217 (AKT3)	[53]
	miR-23a (CCNG1)	[54]
	miR-24	[24]
	miR-26b-5p (Gsk-3β and β-catenin)	[55]
	miR-30a	[24]
	miR-335-5p (ICAM-1)	[56]
	miR-34a	[26]
	miR-34a (MET, XIST)	[57]
	miR-424	[24]
	miR-429 (ZEB1)	[58]
	miR-451	[59]
	miR-577 (SphK2)	[60]
	miR-615	[24]
	miR-9 (BRAF)	[61]
	miR-let-7e (HMGB1)	[62]

**Table 2 genes-14-00685-t002:** Target genes for microRNA 125b.

Target Gene	Description
*ACTN4*	Alpha-actinin-4;
*ARHGAP35*	Rho GTPase-activating protein 35;
*CAV2*	Caveolin-2;
*CCDC6*	Coiled-coil domain-containing protein 6
*CDK6*	Cyclin-dependent kinase 6;
*CDKN1B*	Cyclin-dependent kinase inhibitor 1B;
*COL1A2*	Collagen alpha-2(I) chain;
*COL6A1*	Collagen alpha-1(VI) chain;
*CSNK1E*	Casein kinase I isoform epsilon;
*DOT1L*	Histone-lysine N-methyltransferase;
*ELK1*	ETS domain-containing protein Elk-1;
*FLNA*	Filamin-A;
*FN1*	Fibronectin 1;
*GART*	Phosphoribosylamine--glycine ligase;
*HDAC2*	Histone deacetylase 1/2;
*HGF*	Hepatocyte growth factor;
*HK1*	Hexose kinase.
*HLA-A*	HLA class I histocompatibility antigen;
*IGF2*	Insulin-like growth factor II;
*IL6*	Interleukin-6;
*KMT2A*	[histone h3]-lysine4 n-trimethyltransferase;
*KMT2D*	Histone methyltransferase. Methylates ‘Lys-4’ of histone H3 (H3K4me).
*MAPK1*	Mitogen-activated protein kinase 1;
*MAPK3*	Mitogen-activated protein kinase 3;
*MTHFD1*	C-1-tetrahydrofolate synthase;
*MTHFD1L*	Monofunctional C1-tetrahydrofolate synthase;
*MYC*	Myc proto-oncogene protein;
*OGDH*	2-oxoglutarate dehydrogenase;
*PHYKPL*	5-phosphohydroxy-L-lysine phospho-lyase;
*PIK3CA*	Phosphatidylinositol 4,5-bisphosphate 3-kinase;
*PKM*	Pyruvate kinase m1/2; Pyruvate kinase PKM;
*PLK1*	Serine/threonine-protein kinase PLK1;
*PPP1CB*	Serine/threonine-protein phosphatase PP1-beta catalytic subunit;
*RANBP1*	Ran-specific GTPase-activating protein;
*RBL2*	Rb transcriptional corepressor like 2;
*RHOA*	Ras homolog gene family, member a;
*SETD8*	[histone h4]-lysine20 n-methyltransferase setd8;
*SMAD4*	Mothers against decapentaplegic homolog 4;
*SOD2*	Superoxide dismutase [Mn], mitochondrial;
*TCF7*	Transcription factor 7;
*THBS1*	Thrombospondin-1;
*TNC*	Tenascin;
*UBR4*	E3 ubiquitin-protein ligase UBR4;
*WNT11*	Wingless-type mmtv integration site family, member 11

**Table 3 genes-14-00685-t003:** The signaling cascades and their components, associated with miR-125b.

Signaling Cascade	Components
ECM-receptor interaction	THBS1
	COL6A1
	COL1A2
	FN1
	TNC
	LAMA4
Proteoglycans in cancer	THBS1
	RHOA
	CAV2
	MYC
	IGF2
	FLNA
	MAPK3
	WNT11
	PIK3CA
	FN1
	HGF
	MAPK1
	ELK1
	PPP1CB
Lysine degradation	OGDH
	SETD8
	PHYKPL
	KMT2D
	DOT1L
	KMT2A
Adherens junction	PVRL2
	RHOA
	TJP1
	SMAD4
	MAPK3
	ACTN4
	EP300
	TCF7
	MAPK1
Viral carcinogenesis	RBL2
	RANBP1
	PKM
	CDKN1B
	RHOA
	CDK6
	HDAC2
	MAPK3
	ACTN4
	EP300
	PIK3CA
	MAPK1
	HLA-A
	UBR4
Thyroid cancer	MYC
	MAPK3
	TCF7
	MAPK1
	CCDC6
Focal adhesion	THBS1
	RHOA
	CAV2
	COL6A1
	ARHGAP35
	FLNA
	MAPK3
	COL1A2
	ACTN4
	PIK3CA
	FN1
	TNC
	HGF
	MAPK1
	ELK1
	PPP1CB
	LAMA4
One carbon pool by folate	MTHFD1L
	MTHFD1
	GART
TGF-beta signaling pathway	THBS1
	RHOA
	SMAD4
	MYC
	MAPK3
	EP300
	MAPK1
Colorectal cancer	RHOA
	SMAD4
	MYC
	MAPK3
	PIK3CA
	TCF7
	MAPK1
Central carbon metabolism in cancer	PKM
	MYC
	MAPK3
	PIK3CA
	HK1
	MAPK1
FoxO signaling pathway	RBL2
	CDKN1B
	SMAD4
	MAPK3
	CSNK1E
	EP300
	SOD2
	PIK3CA
	PLK1
	IL6
	MAPK1

**Table 4 genes-14-00685-t004:** The miR-125b expression in tumor tissue in patients with thyroid pathology.

Indicator	Hsa-miR-125-b Expression Level (Relative Units)	
Patients with thyroid pathology	Benign thyroid pathology	6.75 (1.15; 18.38)
	PTC	1.00 (0.19; 4.59) #
Tumor size	T1-2N0M0	1.15 (0.07; 10.56)
	T3-4N0-1M0	1.00 (0.62; 3.03)
Regional metastases	T1-2N0M0	1.07 (0.14; 4.59)
	T3-4N1M0	0.81 (0.19; 18.38)
Invasion	−	1.00 (0.14; 3.03)
	+	2.63 (0.19; 24.25) *
Histological subtype	Classical subtype	1.11 (0.33; 10.56)
	Follicular subtype	0.03 (0; 0.14) ***
BRAFV600E mutational status	−	2.66 (0.87; 32.00)
	+	0.21 (0.01; 0.66) **
Risk of disease recurrence after treatment (ATA)	Low risk	0.11 (0.02; 1.24)
	Intermediate risk	11.78 (2.39; 28.13) ****
	High risk	1.00 (0.62; 1.52)

Note: #—significance of differences compared to patients with benign tumors, *p* < 0.05; *—significance of differences compared to patients with signs of tumor, *p* < 0.05; **—significance of differences compared to patients with follicular subtype of PTC, *p* < 0.05; ***—significance of differences compared to patients with BRAFV600E mutation, *p* < 0.05; ****—significance of differences compared with patients in group with low risk of recurrence after the treatment, *p* < 0.05.

## Data Availability

All the relevant data have been provided in the manuscript, and any Appendix A used and/or analyzed during the current study are available from the corresponding author on reasonable request.

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
