# Peer review of "Autophagy-Related MicroRNA: Tumor miR-125b and Thyroid Cancers"

_genes, 2023, doi:10.3390/genes14030685_

Round 1

Reviewer 1 Report

The present work is attempting to shed light in the role of scepcific miRNAs on thyroid cancer behavior, using data from >100 thyroid surgical specimens, inluding 67 cancers. Before any further review could be performed, two major issues need to be addressed.

1. There are major linguistic issues. Please have a native English speaker of adequate scientific background review the manuscript. 

2. In the results section it is impossible to understand what are the experiments that you performed and what the results were. In every sentence and every table we see explanations of what the different genes/molecules do in each pathway and references to other works. What did you find? Were is the statistical analysis of your findings? All the tables would be useful in the discussion section of the manuscript, or if that manuscript was a review. Please help us understand your important findings...!

Author Response

Response to the Reviewer 1

The Team of the authors are thankful to the Reviewer’s attention to the paper

  1. The paper was modified with aim to clarify all methods applied for the investigation
  2. The extensive proofreading was made with the help of the specialists.

The concept of the work is to create the connections between the theoretical finding and found facts and ideas and experimental findings.

The concept of the paper was to find the pool of miRs associated with the autophagy, then identify the most powerful ones that regulate the oncogenesis in the thyroid gland The interesting trend was again verified for the thyroid cancers, that determination of the one isolated single pathway and rout does not explain and predict the cancer cells behavior. There should be found the molecular factors combination that are essential for the oncogenesis.  They provide the search for the new prominent targets in anti-cancer therapy.

Reviewer 2 Report

Liudmila and collaborators wrote an interesting paper about miR-125b in papillary thyroid cancer. They made a good background research on miRNA 125 regulation products. They suggest that miRNA125 levels could indicate autophagy dysregulation in papillary cancer. However, authors should address the following items.

In line 35, the authors stated, “The key 35 triggers determining the autophagy induction include the cascade of the proteins”. Authors must take into consideration that autophagy is a basal active pathway that can be triggered by several cues, such as starvation, reactive oxygen stress, endoplasmic stress, and hypoxia. Therefore, there are several pathways modulating autophagy, such as AMPK, calcineurin, and PI3K. The authors should rephrase the sentence and make it more integrative.

In line 37, the reference is written as [Holm TM, 2022]. However, the other references are enumerated. Please, correct it.

In line 63, the authors wrote,” The increased expression of p-ULK1 can promote autophagy and inhibit the expression of inflammatory factors [6]”. However, in this work autophagy was inaccurately measured. The research lacks proper controls and inhibitors. The authors are encouraged to take a close look at all the autophagy papers that they cited. Also, the authors are referred to the Guidelines for the use and interpretation of assays for monitoring autophagy.

Klionsky, D. J., Abdel-Aziz, A. K., Abdelfatah, S., Abdellatif, M., Abdoli, A., Abel, S., Abeliovich, H., Abildgaard, M. H., Abudu, Y. P., Acevedo-Arozena, A., Adamopoulos, I. E., Adeli, K., Adolph, T. E., Adornetto, A., Aflaki, E., Agam, G., Agarwal, A., Aggarwal, B. B., Agnello, M., Agostinis, P., … Tong, C. K. (2021). Guidelines for the use and interpretation of assays for monitoring autophagy (4th edition)1Autophagy17(1), 1–382. https://doi.org/10.1080/15548627.2020.1797280

In the paragraph from lines 63 to 66, the authors wrote about three different autophagic-related (Atg) proteins. However, there is no relation or integration between the paragraphs. Please, rephrase. It would be a better idea to briefly discuss the different Atg proteins, their classification, and their function.

The paragraph from lines 82 to 86 needs references. In which types of cancers do miRNAs have been implicated?

In the methodology section, the real-time PCR should be described as the MIQE guidelines. The authors are referred to the proper guideline.

Bustin, S. A., Benes, V., Garson, J. A., Hellemans, J., Huggett, J., Kubista, M., Mueller, R., Nolan, T., Pfaffl, M. W., Shipley, G. L., Vandesompele, J., & Wittwer, C. T. (2009). The MIQE guidelines: minimum information for publication of quantitative real-time PCR experiments. Clinical chemistry55(4), 611–622. https://doi.org/10.1373/clinchem.2008.112797

In line 388, the authors stated, “In previous 388 research, the LC3B protein content, an autophagosome protein, was noted in this group 389 of patients [90]. “. Please discuss further, was this protein increased/decreased? Moreover, autophagy is a dynamic process; when it is measured, it is imperative to use a pharmacological inhibitor (chloroquine, hydroxychloroquine, bafilomycin, Lys05, 3-methyladenine, etc.) and a genetical inhibition to ATG genes. The authors are only measuring in a static assay miR-125 and not the autophagic process per se. Authors should discuss the limitation of their study or how they could approach in a further study the lack of proper inhibitors in solid primary cells and how this affects the interpretation of the autophagic process. Authors are referred to the above Guidelines for the use and interpretation of assays for monitoring autophagy and the following papers.

Mizushima, N., & Yoshimori, T. (2007). How to interpret LC3 immunoblotting. Autophagy3(6), 542–545. https://doi.org/10.4161/auto.4600

Humbert, M., Morán, M., de la Cruz-Ojeda, P., Muntané, J., Wiedmer, T., Apostolova, N., McKenna, S. L., Velasco, G., Balduini, W., Eckhart, L., Janji, B., Sampaio-Marques, B., Ludovico, P., Žerovnik, E., Langer, R., Perren, A., Engedal, N., & Tschan, M. P. (2020). Assessing Autophagy in Archived Tissue or How to Capture Autophagic Flux from a Tissue Snapshot. Biology9(3), 59. https://doi.org/10.3390/biology9030059

Author Response

Response for the Reviewer 2

All corrections were made. The abstract of the paper was modified and rewritten. The Introduction part of the paper was rewritten. The aim of the paper was corrected.

The list of the references was modified. The methods applied in the paper were corrected according to the international recommendations.

The Discussion part of the paper was clarified to make logical explanations for the found facts. In the conclusion, it indicated the significance of  the revealed theoretical and experimental data.

Round 2

Reviewer 1 Report

Excellent work

All issues resolved, but the English language, which requires some changes still.